# Study on Optimization of Damping Performance and Damping Temperature Range of Silicone Rubber by Polyborosiloxane Gel

**DOI:** 10.3390/polym12051196

**Published:** 2020-05-24

**Authors:** Jiang Zhao, Nan Jiang, Dongsheng Zhang, Bobing He, Xian Chen

**Affiliations:** 1Room 602, Yifu Science and Technology Building, Wangjiang Campus, Sichuan University, Chengdu 610065, Sichuan, China; 2017322030054@stu.scu.edu.cn; 2North 112, First Science Building, Wangjiang Campus, Sichuan University, Chengdu 610065, Sichuan, China; jiangnan805@outlook.com (N.J.); 2017222030130@stu.scu.edu.cn (D.Z.)

**Keywords:** shear-hardening gel, damping properties, dynamic covalent bonds

## Abstract

Polyborosiloxane gel (PBS-gel) with shear hardening properties was prepared by cross-linking boric acid and hydroxyl-terminated polydimethylsiloxane through B–O–Si dynamic covalent bonding. The prepared PBS gel was mixed with methyl vinyl silicone rubber (MVQ), and a benzoyl peroxide (BPO) cross-linking agent was added to vulcanize the silicone rubber. At the same time, the gel molecules were co-vulcanizing with MVQ to produce molecular cross-linking. The effects of PBS-gel on the damping properties of silicone rubber were analyzed by dynamic rheological test, Fourier transform infrared spectroscopy and dynamic mechanical analysis. The results demonstrated that the damping performance of MVQ/PBS rubber is greatly improved and the rubber has a tanδ > 0.3 in the range of −25~125 °C. The shear-hardening gel is uniformly dispersed in the system, due to the combined action of covalent bonds and intermolecular forces, which act as an active molecular chain that can efficiently dissipate and transfer energy inside the silicone rubber.

## 1. Introduction

Polymer damping material is a kind of functional material which can convert mechanical vibration into heat dissipation [1,2]. This kind of material can be used to reduce the vibration and noise of all kinds of machinery, and to improve the precision and life of machinery. It has been widely used in the fields of transportation, municipal engineering, high-rise buildings, precision instruments, aerospace, military equipment and so on [3,4]. In fact, the effective high damping materials require a damping coefficient of tanδ > 0.3 at a wide temperature range [5]. However, the effective damping temperature of a general viscoelastic material is mostly near the glass transition temperature (*T*_g_) and the temperature range is narrow [6]. With the increase of temperature, the resonance intensity increases, and its damping coefficient (tanδ) is significantly reduced, which makes it impossible to adapt some special working conditions, such as extreme conditions of high and low temperature alternating [7].

Methyl vinyl silicone rubber (MVQ) is a long straight chain polysiloxane with high molecular weight. Its main chain is composed of Si–O–Si bonds, and its side groups are composed of organic groups. Due to the typical semi-organic and semi-inorganic structure, silicone rubber has both the thermal stability of inorganic polymer, and the flexibility of organic polymer. Therefore, silicone rubber has excellent low temperature resistance and resistance to high temperature performance [8,9]. The glass transition temperature of silicone rubber is relatively low (−70~−110 °C), and the structure of Si–O–Si bond makes its mechanical properties keeping stable in a wide temperature range (−50~200 °C) [10]. The damping property of silicone rubber is mainly due to the dynamic deformation under the action of dynamic stress. The external force response of rubber is divided into the elastic part and the viscous part, and the strain will fall behind the stress. Importantly, the mutual friction between the molecules occurs when the rubber exhibits a cyclical change of stretching and retraction. Moreover, the mechanical energy is dissipated by the running of thermal energy, thereby achieving the effect of vibration and noise reduction [11,12]. However, the damping properties of silicone rubber are lower at room temperature, generally near the *T*_g_ (−129~−70 °C), and the tanδ of the material is generally below 0.1. It is necessary to modify the damping property to meet the requirements [13,14].

The commonly used methods include the blending method, the mutual transfer and network methods and the copolymerization method [15,16]. Adding additives is one of the effective methods to improve the damping properties of silicone rubber. A patent of Kobayashi [17] exhibited a way to enhance the damping properties by adding a kind of silicone powder and a surface-hydrophobic silica powder into the rubber, and finally, an excellent vibration damping performance and long-term storage stable rubber was obtained.

Polyborosiloxanes (PBS), which was invented initially in the search for substitutes of natural rubbers, possess reversible physical cross-links. A material like PBS may be denoted a supramolecular elastomer [18,19]. In all commercial applications, PBS constitutes the viscoelastic matrix, and inorganic/organic fillers are added for different engineering applications or lowering cost [20]. At room temperature, pure PBSs behave elastically under a rapid strain variation, and suffer from brittle fractures. However, on longer time-scales, they flow as a viscous fluid. The fascinating viscoelastic properties made PBSs applicable in education on various deformation processes [21]. The molecular structure of the silicone rubber PBS-gel is similar, and the shear hardening gel itself has a strong energy dissipation effect [22,23]. In our present study, the PBS-gel is blended with silicone rubber to prepare composite silicone rubber. The effects of shear hardening gel on the damping properties of silicone rubber are also investigated.

## 2. Materials and Methods

### 2.1. Materials

Polydimethylsiloxanes (PDMS) with hydroxyl at the ends of the linear chains originated from Zhonglan Chenguang (Chengdu, China), the *M*_w_ of hydroxyl PDMS was chosen to 17.3, 49.2, and 91.4 × 10^4^ g·mol^−1^, respectively. According to the three different molecular weights of the blocked hydroxyl group, low, medium and high, PDMS is labeled as: LPDMS, PDMS, HPDMS. Boron acid and vulcanizing agent of benzoyl peroxide (BPO) were purchased from Chengdu Kelong Chemical Co., Ltd., Chengdu, China. Silicone rubber of XIAMETER RBB-2400-30 and methyl vinyl silicone rubber (MVQ) of RBB2400-30 were bought from Dow Corning (Midland, MI, USA).

### 2.2. Fabrication of LPBS/MVQ, PBS/MVQ, and HPBS/MVQ

Firstly, the PDMS is placed in a vacuum oven at 120 °C for 24 h to remove the water. Then, a certain amount of dried PDMS with accurately molecular weight are incorporated with boric acid, mixed at a temperature of 160 °C for 30 min in HAAKE mixer (Rheomix 3000 OS, Haake Technik GmbH, Vreden, Germany). The desired shear gel was obtained, and designated as LPBS, PBS and HPBS. The silicone rubber and the PBS gel are mixed in HAAKE mixer, followed by added 0.5 wt % of BPO at room temperature. Finally, the obtained rubbers are vulcanized at 120 °C for 20 min, to prepare sheet with thickness of 1 mm, and recorded as LPBS/MVQ, PBS/MVQ, HPBS/MVQ, respectively. The gel content of the samples are 10 wt % LPBS/MVQ, 20 wt % LPBS/MVQ, 30 wt % LPBS/MVQ, 40 wt % LPBS/MVQ, 40 wt % PBS/MVQ, 40 wt % HPBS/MVQ. The fabrication process is shown in Scheme 1.

### 2.3. Characterization

Fourier transform infrared (FTIR) spectra of PBS-gels in the range of 4000–500 cm^−1^ were recorded using a Nicolet 8700 spectrometer (Thermo Scientific Nicolet, Madison, WI, USA) with the resolution of 4 cm^−1^.

Dynamic rheologicalwere analyzed by a plate rheometer HR-2 hybrid rheometer (TA Instruments-Waters L.L.C., Newcastle, DE, USA). The sample needs to be pre-sheared at 25 °C for 30 s to eliminate the shear history, and the tested sample was a cylinder with a diameter of 40 mm and a thickness of 1 mm.

The dynamic thermomechanical analysis (DMA) was used to measure the mechanical properties of viscoelastic materials with time, temperature or frequency. The samples were made into 20 mm × 5 mm × 0.5 mm (length × width × thickness) splines. The dynamic heat mechanical analyzer of Q800 (TA Instruments, Newscastle, DE, USA) with a tensile mode was heated from −80 to 150 °C at 3 °C/min, with an amplitude of 30 μm and a vibration frequency of 1 Hz. The temperature was controlled by a liquid nitrogen cooling device throughout the process.

The composite silicone rubber obtained by blending was observed by scanning electron microscopy (SEM), with the operating conditions of 15 KV. Before observation, the sample was brittle fractured with liquid nitrogen, followed by etching in a tetrahydrofuran solvent for half an hour.

The mechanical properties of the rubber were tested using an electronic universal testing machine (DJS-5010DW, Shuangxu Electronics Co., LTD. Shanghai, China), and the compression set of the rubber material was tested.

The compression set tester used for vulcanized rubber are consists of parallel steel plates, limiters and fasteners. The thermoplastic rubber or foam is measured at a certain compression rate under normal temperature, high temperature and low temperature conditions, and the rubber deformation is measured after a certain compression time. The compression set rate of the sample can be calculated by the following Formula (1):(1)$$C=\frac{T_{0}-T_{n}}{T_{i}-T_{n}}\times 100\%$$
where C is compression set rate, *T*_0_ is the original thickness of the sample, *T_n_* is the pad thickness.

## 3. Results and Discussions

### 3.1. Analysis of PBS-Gel

Figure 1a shows the preparation of PBS-gel. The prepared PBS-gel product was subjected to infrared spectrum analysis and the results are shown in Figure 1b.The absorption peak at 2905–2960 cm^−1^ is attributed to the asymmetric stretching of the methyl group of Si–CH_3_. The strong absorption band at 1245–1275 cm^−1^ is attributed to the bending vibration peak of the Si–CH_3_. The peak at 1100 cm^−1^ is related to the Si–O bond. The strong absorption peaks at 890 cm^−1^ and 860 cm^−1^ indicates the formations of Si–O–B bonds [24]. In Figure 1c, the dynamic rheological properties of the PBS-gel were also tested. It can be observed that the storage modulus of the PBS-gel increased sharply with the increased shear frequency. Moreover, the PBS gel had significant shear hardening phenomenon. It has also been found that the maximum storage modulus (G’ max) increases as the molecular weight of the PBS increases. At low shear frequencies, PBS with lower molecular weights produce higher storage moduli. Conversely, PBS with higher molecular weights has higher storage moduli at high shear frequencies. The largest application area for such shear-hardening gels is personal protection. For example, the development of flexible and lightweight body armor is imminent. A series of bullet-proof garments based on new materials have been developed to resist attack and absorb impact energy [25,26,27,28,29].

The prepared protective material can effectively resist impact, puncture and consume impact energy. Figure 1d is a graph of the loss factor of PBS-gel at a frequency of 1 Hz. With the increases of molecular weight, the loss factor became higher, while the damping properties of the PBS-gel show an opposite law to its shear-hardening properties. The reason for this phenomenon is for the polymer to be more easily hardened by shearing, due to the increase amount of molecular entanglement inside the polymer. However, this molecular entanglement is similar to physical cross-linking, which reduces the number of active molecules in the gel and limits molecular motion when applied by external forces. The LPBS, PBS and HPBS are means low molecular weight polyborosiloxane, medium molecular weight polyborosiloxane and high molecular weight polyborosiloxane, respectively. As a result, the energy consumption of molecular friction is reduced [30,31,32,33]. Moreover, the loss factor of the PBS-gel is higher in the range of 50 to 150 °C, which means that the PBS-gel has the potential to enhance the damping properties of the silicone rubber. Furtherore, the Loss modulus (G’) and complex viscosity of LPBS, PBS and HPBS samples at different frequencies are shown in Figure 2. With the increase of shear frequency the loss modulus and complex viscosity of PBS gels with different molecular weights show the same trend. However, there is a significant difference in their loss modulus values. In addition, the loss modulus and complex visicosity of gel increase with the increase of molecular weight.

### 3.2. Damping Properties of Composite Silicone Rubber

The cross-sectional SEM images of silicone rubber and silicone rubber composite etched in tetrahydrofuran for half an hour are shown in Figure 3. Generally speaking, the gel can easily etch away within ten minutes in THF solution. However, the composite material shows no significant difference before and after THF etching. The micro-morphology of the materials before and after compounding are close, and the PBS-gel is evenly fused with silicone rubber. These two materials have good compatibility attributed to the similar molecular structure of PBS-gel and silicone rubber. Another remarkable feature is that, through the connection of the vulcanizing agent, it can be effectively combined together through chemical bonds. The low molecular flexibility and shear hardening of the gel can significantly improve the damping performance of the silicone rubber material. In addition, PBS-gels with different molecular weights have a smaller effect on the compatibility of the system, and can achieve microscopic fusion.

As shown in Figure 4, the silicone rubber vulcanized by BPO has a loss factor of less than 0.15. The loss factor of silicone rubber increased significantly when LPBS-gel was added, and the loss factor peak shifted to the normal temperature range. The loss factor was greater than 0.3, and the temperature range exceeded 125 °C when the gel content exceeded 30 wt %. The inner molecule frictions between the hydrophilic group and the hydrophobic group, the interaction between molecules and the energy dissipation effect are both strong. Moreover, the Si–O–B bond is a reversible covalent cross-linking bond. The relative movement of the molecules, and the constant breakage and reconstruction of Si–O–B will help increase energy consumption when the material is subjected to external forces. In addition, the PBS-gel materials are highly deformable, but they harden under shear, similar to harder solids: this facilitates the diffusion of energy from the local to the whole, thereby increasing energy consumption [34]. As the gel content increases, the loss factor of silicone rubber increases sharply and the temperature domain broadens. This is mainly because when BPO is used as a crosslinking agent, the PBS-gel will be partially crosslinked with silicone rubber. The energy dissipation and shear hardening properties of the gel molecules are used to achieve energy dissipation and enhanced damping performance.

At this time, the modulus of the silicone rubber and the composite silicone rubber is as shown in Figure 5. As the gel content increases, the storage modulus of the composite silicone rubber decreases. This is because the presence of PBS-gel increases the spacing between the silicone rubber molecules and the degree of crosslinking decreases accordingly. The storage modulus is almost constant at high temperatures (above the *T*_g_ zone). With the temperature increases, a rubber platform can be observed. Obviously, at lower temperatures, the addition of PBS-gel reduces the stiffness of the composite silicone rubber, where the presence of the gel increases the mobility of the polymer chain. As the temperature increases, the storage modulus decreases, indicating that the flexibility of the rubber is improved. The storage modulus curves of these composites gradually change toward high temperature, with only one transition. This phenomenon indicates that no phase separation occurs in the polymer structure. The insert curve in the Figure 5 is the loss modulus curve of different composite silicone rubbers. It can be seen that, with the addition of PBS gel, the loss modulus of silicone rubber at low temperatures tends to decrease. Since the gel achieves a good degree of dispersion in the silicone rubber, the addition of the gel reduces the elastic properties and friction between the polymer matrix. Therefore, the loss modulus decreases as the content increases, and the hydrogen bond relaxation inside the gel causes the increased loss factor in silicone rubber.

The loss factors of composite silicone rubber with different molecular weight of PBS-gels are shown in Figure 6. It can be seen that the loss factor peak and the temperature range of the composite silicone rubber decreases when the molecular weight of the gel increases. This is because the lower the molecular weight of the PBS, the shorter the molecular chain length, the relative movement in the silicone rubber, the increase of internal energy consumption, resulting in an increase in the loss factor of the silicone rubber [35]. In contrast, the higher molecular weight and the more pronounced entanglement between molecules increase the physical cross-linking between molecules, which limit the molecular motion, and are not good for improving the damping performance of silicone rubber.

Figure 7 is a curve showing the effect of different molecular weight PBS-gels on the modulus of a composite silicone rubber. The storage modulus and loss modulus of the composite silicone rubber increase when the molecular weight of the gel increases. [36]. This is because the increase in the viscosity of the PBS-gel means that the molecular chain length of the PBS increases. The interaction between two molecules increases, which increasing physical crosslinks. At the same time, due to the increase in cross-linking effect, the loss factor of the composite silicone rubber is rather reduced.

According to the ISO standard [37], for the vibration-isolating rubber bearing, the loss factor tan δ value is required to be higher than 0.1 (equivalent damping ratio 10%) in the frequency range of 0.2 to 5 Hz, and the strain range of 10 to 100%. Figure 8a–d shows the change in tanδ and dynamic frequency of silicone rubber. When the content of LPBS-gel is more than 20%, the loss factor of the composite silicone rubber is greater than 0.1. The larger the molecular weight of the gel, the larger the loss factor under the frequency and strain scanning conditions. These results indicate that the composite silicone rubber has good damping properties, and can be used as a vibration-isolating rubber bearing.

### 3.3. Mechanical Properties of Composite Silicone Rubber

Comparing the compression set ratio of silicone rubber, in Table 1, it can be found that the compression set rate will increase with the gel content of LPBS increases at 25 °C. The reason is that the obtained PBS-gel is continuously static under the action of stress and exhibit strong fluidity. After mixing with silicone rubber, the spacing of the molecular chain of silicone rubber increases, and the internal barrier effect decreases. Therefore, the compression set rate of silicone rubber increases with the gel content. The longer the compression time, the deformation rate is large. The compression set rate decreases when adding the same amount of gel in the silicone rubber, because the PBS-gel increases. The gel and the silicone rubber molecular chain are easily intertwined when the viscosity of the PBS-gel increases. The increase of rubber internal friction hinders the movement of the molecules and enhance the compression permanent denaturation rate of the silicone rubber. Similarly, we obtained a series of experimental results: at a temperature of 75 °C, as the gel content increases, the compression set rate of the silicone rubber increases. An increase in the viscosity of the polydimethylsiloxane will increase the rubber deformation rate. As the compression time increases, the deformation rate of the rubber also increases. In addition, the thermal motion of the molecules increases with the increases of temperature. At this time, a directional pressure is applied to the rubber, and the compression set of the silicone rubber increases.

The effects of PBS-gel on tensile strength (TS) and elongation at break (EB) of silicone rubber are shown in Table 2. The TS of the two-component LPBS/MVQ composite is slightly lower than that of pure silicone rubber. Although PBS-gel and silicone rubber can be well mixed, peroxy compounds will cause the appearance of cross-linked structures. However, due to the low molecular weight of polyborosiloxane gel and the weak mechanical properties of the gel material, the related mechanical properties of the PBS/MVQ composite rubber will decrease, and the tensile modulus of the PBS/MVQ composite rubber decreases as the PBS-gel content increases. The EB of PBS/MVQ composite rubber increases with the increase of the content of PBS-gel. The tensile test was carried out at a tensile rate of 50 mm/min. Under the action of this slow and continuous force, gel deformation will occur and make silicone rubber more susceptible to slippage. The PBS shear-hardening gel is a soft gel with strong fluidity under normal temperature and pressure. It acts as a soft phase in the crosslinked network of silicone rubber and has very good soft deformation capacity, so the hardness of PBS/MVQ composite silicone rubber decreases as the PBS-gel content increases.

The tensile modulus and tensile strength of the composite silicone rubber increase with the increase of molecular weight of the PBS, and the molecular weight of the long-chain molecules of the polymer and the vinyl dimethyl silane are increased, so that the physical interaction of the entire rubber system is increased. The joint strength is increased, so the tensile strength of HPBS/MVQ rubber is greater than PBS/MVQ, and is greater than the strength of LPBS/MVQ composite silicone rubber, both of which are greater than 5 MPa. At this time, the EB of rubber decreases due to the increase of gel viscosity, and the hardness of the material also increases accordingly.

The tensile tester stretches the sample to 100%, 200% or 300% and then retracts the un-stretched sample at the same rate. The energy density (*h*) of the hysteresis curve indicates the dissipative energy under cyclic deformation. The strain energy density (*w*) indicates the amounts of undissipated energy under cyclic deformation are shown in Figure 9. The dissipation coefficient (DE) is defined as the ability to dissipate energy, and Formula (2) is:(2)DE=h/(h+w)×100%

The dissipation coefficient of the modified silicone rubber is shown in Table 3. As the strain increases from 100% to 300%, the energy dissipation rate of the material increases, mainly due to the large deformation of the damping silicone rubber. The molecular motion cannot keep up with the change of external force, and the hysteresis is obvious [38]. The trend in strain 300% is different from lower strain, which is due to the greater energy dissipation of the molecular chain of the PBS gel when the strain is large. Moreover, the rubber in the materials may reach the limit of deformation and lose energy when the molecule is highly stretched. When the strain is increase lager, the energy dissipation is more due to the motion of the molecular chain of the PBS gel and large deformation of silicone rubber. As a rubber bearing material, a high DE is indispensable, because the vibration-absorbing energy consumed by the higher composite material when the vibration-isolating rubber is subjected to the equivalent vibration energy. The higher the gel content, the greater the energy dissipation rate of the composite silicone rubber, mainly because the higher the gel content, the more the gel molecules can be effectively dispersed within the material, and the energy dispersibility is improved. The increase of the molecular weight of the gel will reduce the energy dissipation rate of the silicone rubber, mainly because, as the molecular weight of the gel increases, the molecular cross-linking and entanglement between the gel and the silicone rubber increase. The molecular mobility decreases, and the hysteresis effect becomes more obvious, so that the energy dissipation coefficient DE decreases, which is also consistent with the change of the loss angle of the compound silicone rubber.

## 4. Conclusions

In this paper, the effect of polyborosilicone (PBS) gel on the damping properties of methyl vinyl silicone rubber (MVQ) was investigated. The PBS-gel has excellent damping performance and is similar to the molecular structure of silicone rubber, which can achieve good mixing and greatly enhance the damping performance of silicone rubber. The high loss factor is maintained in a wide temperature range, when the contents of LPBS gel reached above 30 wt %, the loss factor of silicone rubber can be more than 0.5, and its effective temperature ranges from −20–125 °C. This is mainly due to the rate dependence of the of the gel, the local energy can quickly transfer local energy to other parts of the rubber, thereby increasing the ability to dissipate energy. In addition, the interaction forces existing between gel molecules and silicone rubber molecules, and the rupture and reconstruction of the hydrogen bonds and the reversible covalent bonds, have caused a variety of relative movements between the molecules, increasing the energy dissipation and energy conversion capabilities. It was found that the loss factor of MVQ was reduced with the increase of molecular weight of the gel, mainly because the size of the long chain molecules directly affected the entanglement between molecules, and the increase of physical crosslinking would reduce the molecular motion, which was not conducive to energy dissipation.

The damping performance and other mechanical properties of the material can be balanced by adjusting the content and the molecular weight of the gel, to meet the performance requirements of different products. It should be noted that the high-damping composites prepared in this paper can continue to be combined with nano-functional particles, such as graphene, piezoelectric particles and carbonyl iron, thereby further improving the application of silicone rubber in the fields of controllable vibration reduction of precision instruments and flexible intelligent devices.

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
