# Peer review of "Study on Optimization of Damping Performance and Damping Temperature Range of Silicone Rubber by Polyborosiloxane Gel"

_polymers, 2020, doi:10.3390/polym12051196_

Round 1

Reviewer 1 Report

The work entitled “Study on Optimization of Damping Performance and Damping Temperature Range of Silicone Rubber byPolyborosiloxane Gel” is a good one, but its quality needs to be enhanced before publication.

Comments:

  1. The work needs a comprehensive investigation of structure-property relationships. So please clarify any trend you observed.
  2. Please report gel content of all samples.
  3. Introduction is poor and it has to be improved by a thorough clarification of the prior works and related innovations as well as citing more references.
  4. Please define MVQ in materials section.
  5. Fig 1 c: Please report all rheological data including complex viscosity and loss modulus curves for the samples (LPBS with different contents as well as PBS with different molecular weights) .
  6. “the silicone rubber vulcanized by BPO has a loss factor of less than 1.5”. It should be 0.15 not 1.5.
  7. table 1: by comparing the data for 10 wt.% LPBS/MVQ and 20 wt.% LPBS/MVQ samples it can be found that there is a decrease by increasing LPBS content. Please justify.

  1. Dissipation coefficient: The trend in strain 300% is completely different from that is obserbed in lower strains. Why? Please justify and cite refs.
  2. Conclusions should be re-written. Please add some important trends to conclusions.
  3. Based on the results, the authors must mention the best sample for the corresponding application.
  4. The English is very poor. The manuscript is full of mistakes, etc. so it must be re-written by a native speaker. Some of the cases are as follows:

-were synthesis

-the moisture water?!

-with accurately weigh

-followed by mixed

-With the molecular weight increases

-SEMs image

-attribute to the similar

-The mainly reason

-molecule friction

-polymer matrixes

-The tensile test is carried out

-the gel itself Deformation

-with the increase molecular weight

-It should be note

Author Response

Reviewer: 1

Comment 1:

 “The work entitled “Study on Optimization of Damping Performance and Damping Temperature Range of Silicone Rubber by Polyborosiloxane Gel” is a good one, but its quality needs to be enhanced before publication.”

Response 1:

We acknowledge the reviewer’s comments.

Comment 2:

“1. The work needs a comprehensive investigation of structure-property relationships. So please clarify any trend you observed.”

Response 2:

As pointed out by the reviewer, the structure of the polymer material determines the properties. With the molecular increases of weight, the loss factor became higher, while the damping properties of the PBS-gel show an opposite law to its shear-hardening properties. The inner molecule frictions between the hydrophilic group and the hydrophobic group, the interaction between molecules and the energy dissipation effect are both strong. Moreover, the Si-O-B bond is a reversible covalent cross-linking bond. The relative movement of the molecules, and the constant breakage and reconstruction of Si-O-B will help increase energy consumption when the material is subjected to external forces. The storage modulus and loss modulus of the composite silicone rubber increase when the molecular weight of the gel increases.

According to the reviewer’s suggestion, some sentences have been added in the revised manuscript.

On Page 4, in the second paragraph of Section 3.1:

With the increases of molecular weight, the loss factor became higher; while the damping properties of the PBS-gel show an opposite law to its shear-hardening properties. The reason of this phenomenon is the polymer to be more easily hardened by shearing due to the increase amount of molecular entanglement inside the polymer.

On Page 6, in the second paragraph of Section 3.2:

The inner molecules friction between the hydrophilic group and the hydrophobic group, the interaction between molecules and the energy dissipation effect are both strong. Moreover, the Si-O-B bond is a reversible covalent cross-linking bond. The relative movement of the molecules, and the constant breakage and reconstruction of Si-O-B will help increase energy consumption when the material is subjected to external forces.

On Page 8, in the fourth paragraph of Section 3.2:

The storage modulus and loss modulus of the composite silicone rubber increase when the molecular weight of the gel increases.

Comment 3:

“2. Please report gel content of all samples.”

Response 3:

  According to the reviewer’s suggestion, the gel content of all has been defined in the section 2.2.

On Page 2, in the Section 2.2:

The gel content of the samples are 10 wt.% LPBS/MVQ, 20 wt.% LPBS/MVQ, 30 wt.% LPBS/MVQ, 40 wt.% LPBS/MVQ, 40 wt.% PBS/MVQ, 40 wt.% HPBS/MVQ.

Comment 4:

“3. Introduction is poor and it has to be improved by a thorough clarification of the prior works and related innovations as well as citing more references.”

Response 4:

The reviewer’s question is significant.

According to the reviewer’s suggestion, we have through clarified the prior works in the revised manuscript.

In addition, some new references have also been added in the manuscript.

Ref. 6. Sperling, L.; Fay, J. Factors which affect the glass transition and damping capability of polymers. Polymers for Advanced Technologies, Polym. Advan. Technol. 1991, 2, 49-56.

Ref. 17. Tang, M.; Wang, W.; Xu, D.; Wang, Z. Synthesis of structure-controlled polyborosiloxanes and investigation on their viscoelastic response to molecular mass of polydimethylsiloxane triggered by both chemical and physical interactions, Ind. Eng. Chem. Res. 2016, 55, 12582-12589.

Ref. 30. Plant, D.; Leevers, P. Mechanical and rheological testing to develop thermoplastic elastomer-polyborodimethylsiloxane blends for personal impact protection, Polym. Test. 2020, 106477.

Comment 5:

“4. Please define MVQ in materials section.”

Response 5:

  According to the reviewer’s suggestion, methyl vinyl silicone rubber (MVQ) has been defined in materials section.

On Page 2, in the Section 2.1:

Silicone rubber of XIAMETER RBB-2400-30 and methyl vinyl silicone rubber (MVQ) of RBB2400-30 were bought from Dow Corning.

Comment 6:

“5. Fig 1 c: Please report all rheological data including complex viscosity and loss modulus curves for the samples (LPBS with different contents as well as PBS with different molecular weights).”

Response 6:

The reviewer’s question is significant.

According to the reviewer’s suggestion, the rheological data have been added in the revised manuscript.

On Page 4, in the Section 3.1:

Figure 2. (a) LPBS with different contents; (b) PBS with different molecular weights.

Comment 7:

“6. “the silicone rubber vulcanized by BPO has a loss factor of less than 1.5”. It should be 0.15 not 1.5.”

Response 7:

Yes, this is our mistake.

The loss factor the 1.5 has been revised to 0.15.

Comment 8:

“7. table 1: by comparing the data for 10 wt.% LPBS/MVQ and 20 wt.% LPBS/MVQ samples it can be found that there is a decrease by increasing LPBS content. Please justify.”

Response 8:

Yes, this is our mistake.

The comparing data of deformation rate of silicone rubber with different gel content has been corrected in the revised manuscript.

Comment 9:

“8. Dissipation coefficient: The trend in strain 300% is completely different from that is obserbed in lower strains. Why? Please justify and cite refs.”

Response 9:

  The energy dissipation rate of the material attribute to the large deformation of the damping silicone rubber. The trend in strain 300% is different from lower strain is due to the more energy dissipation of the molecular chain of the PBS gel when the strain is large. Moreover, the rubber in the materials may reach the limit of deformation and lose energy when the molecule is highly stretched. So, the trend in strain 300% is different from lower strain of 100% and 200%.

In addition, some sentences have been added in the revised manuscript.

On Page 10, in the fourth paragraph of Section 3.4:

The trend in strain 300% is different from lower strain is due to the more energy dissipation of the molecular chain of the PBS gel when the strain is large. Moreover, the rubber in the materials may reach the limit of deformation and lose energy when the molecule is highly stretched. When the strain is increase lager, the energy dissipation is more due to the motion of the molecular chain of the PBS gel and large deformation of silicone rubber.

Comment 10:

“9. Conclusions should be re-written. Please add some important trends to conclusions.”

Response 10:

The reviewer’s suggestion is significant.

According to the reviewer’s suggestion, the Conclusion has been re-written in the revised manuscript.

On page 11, in the Section 4.

In this paper, the effect of polyborosilicone (PBS) gel on the damping properties of methyl vinyl silicone rubber (MVQ) was investigated. The PBS-gel has excellent damping performance and similar to the molecular structure of silicone rubber, which can achieve good mixing and greatly enhance the damping performance of silicone rubber, especially maintain a high loss factor in a wide temperature range, when the contents of LPBS gel reached above 30 wt%, the loss factor of silicone rubber can be more than 0.5, and its effective temperature range from -20-125 ℃. This is mainly due to rate dependence of the gel can quickly transfer local energy to other parts of the rubber, thereby increasing the ability to dissipate energy. In addition, the interaction forces existing between gel molecules and silicone rubber molecules and the rupture and reconstruction of hydrogen bonds and reversible covalent bonds has caused a variety of relative movements between the molecules, increasing the energy dissipation and energy conversion capabilities. It was found that the loss factor of MVQ was reduced with the increase of molecular weight of the gel, mainly because the size of the long chain molecules directly affected the entanglement between molecules, and the increase of physical crosslinking would reduce the molecular motion, which was not conducive to energy dissipation.

The damping performance and other mechanical properties of the material can be balanced by adjusting the content and molecular weight of the gel to meet the performance requirements of different products. It should be note that the high-damping composites prepared in this paper can continue to be combined with nano-functional particles, such as graphene, piezoelectric particles and carbonyl iron, thereby further improving the application of silicone rubber in the fields of controllable vibration reduction of precision instruments and flexible intelligent devices.

Comment 11:

“10. Based on the results, the authors must mention the best sample for the corresponding application.”

Response 11:

The reviewer’s suggestion is significant.

Based on the results, the damping performance of silicon rubber was the best when the contents of LPBS gel reached 40 wt%. The loss factor of the sample could reach 0.5 in the temperature range of 20-125 ℃. It should be noted that the high-damping composites prepared in this paper can continue to be combined with nano-functional particles, such as graphene, piezoelectric particles and carbonyl iron, thereby further improving the application of silicone rubber in the fields of controllable vibration reduction of precision instruments and flexible intelligent devices.

In addition, some sentenceS have been added in the revised manuscript.

On page 11, in the Section 4.

It should be noted that the high-damping composites prepared in this paper can continue to be combined with nano-functional particles, such as graphene, piezoelectric particles and carbonyl iron, thereby further improving the application of silicone rubber in the fields of controllable vibration reduction of precision instruments and flexible intelligent devices.

Comment 12:

“The English is very poor. The manuscript is full of mistakes, etc. so it must be re-written by a native speaker. Some of the cases are as follows:

-were synthesis

-the moisture water?!

-with accurately weigh

-followed by mixed

-With the molecular weight increases

-SEMs image

-attribute to the similar

-The mainly reason

-molecule friction

-polymer matrixes

-The tensile test is carried out

-the gel itself Deformation

-with the increase molecular weight

-It should be note”

Response 12:

  The reviewer’s suggestion is significant.

  The English syntax has been revised in the revised manuscript.

Reviewer 2 Report

In my opinion this manuscript should be revised by authors.

My detailed comments in the attachment.

Author Response

Reviewer: 2

Comment 1:

Add full affiliations and addresses.

Response 1:

Jiang Zhao 1, PhD , College of Chemistry, Sichuan University.

Address:Room 602, Yifu Science and Technology Building, Wangjiang Campus, Sichuan University, Chengdu, Sichuan, China.

Nan Jiang 2, Master degree, College of Chemistry, Sichuan University.

Address:North 112, First Science Building, Wangjiang Campus, Sichuan University, Chengdu, Sichuan, China.

Dongsheng Zhang 3, Master degree, College of Chemistry, Sichuan University.

Address:North 112, First Science Building, Wangjiang Campus, Sichuan University, Chengdu, Sichuan, China.

Bobing He 1*,Associate Professor, College of Chemistry, Sichuan University.

Address:North 112, First Science Building, Wangjiang Campus, Sichuan University, Chengdu, Sichuan, China.

Xian Chen2*, Associate Professor, College of Chemistry, Sichuan University.

Address:Room 602, Yifu Science and Technology Building, Wangjiang Campus, Sichuan University, Chengdu, Sichuan, China.

Comment 2:

In page 2, line 62, I cannot found in the manuscript, information about application of different PDMS. Could you explain?

Response 2:

The reviewer’s question is significant.

The information of different PDMS has been added in the Section 2.1

PDMS with hydroxyl at the ends of the linear chains originated from Zhonglan Chenguang,the Mw of hydroxyl PDMS was chosen to 17.3, 49.2, and 91.4 kg·mol−1, respectively.

Comment 3:

In Figure 2, define meaning of this numbers

Response 3:

  According to the reviewer’s suggestion, the Figure 2 has been revised in the revised manuscript.

Comment 4:

In Table 1, round to decimal all values.

Response 4:

  According to the reviewer’s suggestion, the value in Table 1 has been round to decimal.

Comment 5:

In Table 2, please provide standard deviation values for these values.

Response 5:

  According to the reviewer’s suggestion, the standard deviation values have been added in Table 2 in the revised manuscript.

Comment 5:

In Figure 6, “PBS/MVQ composite rubber”, mention which type of compound was evaluated

Response 5:

The reviewer’s question is significant.

In fact, this is a representative diagram of cyclic stress-strain curves of all the PBS/MVQ, and does not refer to a particular sample, and the final calculation results have been shown in table 2.

Comment 6:

In page 11, line 288, “This is mainly due to the synergistic effect of various interaction forces existing between gel molecules and silicone rubber molecules.” Provide some clear evidence based on the results.

Response 6:

The reviewer’s question is significant.

It is mentioned in this paper that the viscoelastic relaxation of hydroxyl-terminated polysiloxane makes the crosslinked network have good damping performance over a wide temperature range. The same applies to PBS/MVQ, in silicone rubber; the relaxation of polyborosiloxane molecules also increases energy dissipation by rubbing against cross-linked silicone rubber molecules.

Comment 7:

Check reference style according to guidance for authors.

Response 7:

  The correct reference style has been revised in the revised manuscript.

Reviewer 3 Report

This manuscript investigated the effect of PBS-Gel on the damping properties of silicone rubber. The PBS-Gel has excellent damping performance and similar to the molecular structure of silicone rubber, which can achieve good mixing. So PBS-Gel can greatly enhance the damping performance of silicone rubber, especially maintain a high loss factor in a wide temperature range, which proves that the damping performance of the material can adapt to a wide temperature range. However, the experimental results do not fully support this conclusion. Overall, I suggest that this article should be rejected. My detailed comments are as follow:

(1) In figure 1d, the tan δ of pure PBS-Gel is greater than 1, but when it is compounded with silicone rubber(even wt%=40%), the tan δ of composite material is only 0.5, so I do not think PBS-Gel can effectively enhance the damping performance of silicone rubber.

(2) In figure 1, the dynamic bond of B-O-Si should be marked and the first appearance of LPBS and HPBS needs to be explained.

(3) In figure 2, the scale bar should be given more conspicuously/

(4) It is suggested to first discuss the effect of PBS-Gel with different molecular weight on the damping performance of the composite, and then discuss the effect of the added amount of PBS-Gel on the damping performance of the composite.

(5) There are some typos in the article, such as in page 5,line 137;page 6, line 151 and page 7, line 185.

Author Response

Reviewer: 3

Comment 1:

“This manuscript investigated the effect of PBS-Gel on the damping properties of silicone rubber. The PBS-Gel has excellent damping performance and similar to the molecular structure of silicone rubber, which can achieve good mixing. So PBS-Gel can greatly enhance the damping performance of silicone rubber, especially maintain a high loss factor in a wide temperature range, which proves that the damping performance of the material can adapt to a wide temperature range. However, the experimental results do not fully support this conclusion. Overall, I suggest that this article should be rejected. My detailed comments are as follow:”

Response 1:

We acknowledge the reviewer’s comments.

Comment 2:

“(1) In figure 1d, the tan δ of pure PBS-Gel is greater than 1, but when it is compounded with silicone rubber (even wt%=40%), the tan δ of composite material is only 0.5, so I do not think PBS-Gel can effectively enhance the damping performance of silicone rubber.”

Response 2:

  The reviewer’s question is significant.

In fact, it was because of the excellent danping properties of PBS-gel that we compounded it with silicone rubber. Polyborosilone gel (PBS-gel) is a kind of polymer with excellent damping properties and fluidity, which is difficult to be applied directly due to its high fluidity. Methyvinyl silicone rubber(MVQ), is stable in chemical properties and has a but poor damping performance. Because polyborosiloxane and silicone rubber have similar molecular structure, the damping performance of silicone rubber can be improved by adding a small amount of PBS gel to expand the application in the field of damping materials.

Comment 3:

“(2) In figure 1, the dynamic bond of B-O-Si should be marked and the first appearance of LPBS and HPBS needs to be explained.”

Response 3:

The reviewer’s suggestion and question are significant.

According to the reviewer’s suggestion, the dynamic bond of B-O-Si is marked in the figure 1a.

Moreover, the LPBS, PBS, and HPBS are means low molecular weight polyborosiloxane, medium molecular weight polyborosiloxane and high molecular weight polyborosiloxane, respectively.

In addition, some sentences have been added in the revised manuscript.

On page, in the second paragraph of Section on 3.1

The LPBS, PBS, and HPBS are means low molecular weight polyborosiloxane, medium molecular weight polyborosiloxane and high molecular weight polyborosiloxane, respectively.

Comment 4:

“(3) In figure 2, the scale bar should be given more conspicuously/”

Response 4:

The reviewer’s suggestion is significant.

The scale bar has been clearly marked in Figure 3.

Comment 5:

“(4) It is suggested to first discuss the effect of PBS-Gel with different molecular weight on the damping performance of the composite, and then discuss the effect of the added amount of PBS-Gel on the damping performance of the composite.”

Response 5:

The reviewer’s suggestion is significant.

According to the reviewer’s suggestion, the effect of PBS-Gel with different molecular weight on the damping performance of the composite has been first discussed. Furthermore, the effect of the added amount of PBS-Gel on the damping performance of the composite was also discussed.

In addition, Figure 8 has been revised in the revised Manuscript.

Figure 8. (a) and (b) are the loss factors graph with changes in strain of LPBS, PBS, and HPBS compound silicone rubber at different frequencies. (c) and (d) are the graphs of the loss factors of LPBS-Gel compound silicone rubber with different contents as a function of frequency and strain.

Comment 6:

“(5) There are some typos in the article, such as in page 5, line 137; page 6, line 151 and page 7, line 185.”

Response 6:

Yes, this is our mistake.

The typos in the article have been revised in the revised manuscript.

Reviewer 4 Report

I think this article has an adequate amount of data and experiments that may be interesting for some readers, but presents serious flaws that need to be improved before publication:

  • English: even being my self a non-native english speaker I can easily detect many grammatical mistakes and also inapropiate sentences, expressions etc.
  • References are numbered twice (?)
  • Structure:the division of the main points of the paper is not very clear. it makes difficult the understanding. I. e. Point: "analysis PBS-Gel" In my opinion, every figure in this point should have its own caption and number and be explained in a separate pearagraph. The same idea should be used in the rest of the paper.
  • References: in some cases I think references are not really significant of the content. For instance in this same paragraph  "The strong absorption peaks at 890 cm-1 and 860 cm-1 indicates the  formations of Si-O-B bonds [16]"  even if I agree with the band assignation to the I think that the provided reference is not correct. Authors should check these cases and also e careful of providing specific references to specific affirmations not only in this case but along the whole paper.
  • there is an unusual way of including and blending main  concepts related the generalities of material and applications with the specific scientific discussion,.
  • I think the pictures of the materials obtained by SEM do not support only by themselves the provided discussion. By SEM it is difficult to observe the phenmonena described by authors
  • Figures 3a and 3b should also be separated (as said before) and discussion lacks references and/or further experimentation. Especially the allusions to the dynamic bonding.

These same ideas can be applied to other parts of the paper.  I suggest a thoroughly and complete revision of the document.

Author Response

Reviewer: 4

Comment 1:

“I think this article has an adequate amount of data and experiments that may be interesting for some readers, but presents serious flaws that need to be improved before publication:”

Response 1:

We acknowledge the reviewer’s comments.

Comment 2:

“English: even being myself a non-native english speaker I can easily detect many grammatical mistakes and also inapropiate sentences, expressions etc.”

Response 2:

The reviewer’s suggestion is significant.

The English syntax has been revised in the revised manuscript.

Comment 3:

“References are numbered twice (?)”

Response 3:

Yes, this is our mistake.

The correct references have been revised in the revised manuscript.

Comment 4:

“Structure: the division of the main points of the paper is not very clear. it makes difficult the understanding. I. e. Point: "analysis PBS-Gel" In my opinion, every figure in this point should have its own caption and number and be explained in a separate paragraph. The same idea should be used in the rest of the paper.”

Response 4:

  The reviewer’s suggestion is significant.

We have adjusted the structure of the article and further explained the datas.

The Figure 3 has been separated to Figure 4 and Figure 5. The Figure 4 has been separated to Figure 6 and Figure 7.

Comment 5:

“References: in some cases I think references are not really significant of the content. For instance in this same paragraph "The strong absorption peaks at 890 cm-1 and 860 cm-1 indicates the formations of Si-O-B bonds [16]" even if I agree with the band assignation to the I think that the provided reference is not correct. Authors should check these cases and also a careful of providing specific references to specific affirmations not only in this case but along the whole paper.”

Response 5:

The reviewer’s suggestions are absolutely right.

The references have been replaced, and other references in the article have been supplemented and confirmed in the revised manuscript.

Comment 6:

“There is an unusual way of including and blending main concepts related the generalities of material and applications with the specific scientific discussion.”

Response 6:

The reviewer’s suggestions are absolutely right.

According to the reviewer’s suggestion, the main concepts related the generalities of material and applications have been added and revised in the revised manuscript.

On page 11, in the Section 4.

In this paper, the effect of polyborosilicone (PBS) gel on the damping properties of methyl vinyl silicone rubber (MVQ) was investigated. The PBS-gel has excellent damping performance and similar to the molecular structure of silicone rubber, which can achieve good mixing and greatly enhance the damping performance of silicone rubber, especially maintain a high loss factor in a wide temperature range, when the contents of LPBS gel reached above 30 wt%, the loss factor of silicone rubber can be more than 0.5, and its effective temperature range from -20-125 ℃. This is mainly due to rate dependence of the gel can quickly transfer local energy to other parts of the rubber, thereby increasing the ability to dissipate energy. In addition, the interaction forces existing between gel molecules and silicone rubber molecules and the rupture and reconstruction of hydrogen bonds and reversible covalent bonds has caused a variety of relative movements between the molecules, increasing the energy dissipation and energy conversion capabilities. It was found that the loss factor of MVQ was reduced with the increase of molecular weight of the gel, mainly because the size of the long chain molecules directly affected the entanglement between molecules, and the increase of physical crosslinking would reduce the molecular motion, which was not conducive to energy dissipation.

The damping performance and other mechanical properties of the material can be balanced by adjusting the content and molecular weight of the gel to meet the performance requirements of different products. It should be note that the high-damping composites prepared in this paper can continue to be combined with nano-functional particles, such as graphene, piezoelectric particles and carbonyl iron, thereby further improving the application of silicone rubber in the fields of controllable vibration reduction of precision instruments and flexible intelligent devices.

Comment 7:

“I think the pictures of the materials obtained by SEM do not support only by themselves the provided discussion. By SEM it is difficult to observe the phenmonena described by authors”

Response 7:

  The reviewer’s suggestion is significant.

As can be seen from the scanning electron microscope, there is no obvious boundary between the two. This is because the structure of the gel is similar to that of silicon rubber, which can be well mixed.

Comment 8:

“Figures 3a and 3b should also be separated (as said before) and discussion lacks references and/or further experimentation. Especially the allusions to the dynamic bonding.”

Response 8:

The reviewer’s suggestions are absolutely right.

According to the reviewer’s suggestion, the Figure 3 has been separated and discussion.

Fig 3 a and b take temperature as the variable, and the change of loss factor of PBS/MVQ. By comparing the influence of PBS gel on the loss factor of silicone rubber and the change of energy storage modulus, it is easy to blur the focus when discussed separately.

Round 2

Reviewer 1 Report

The authors haven’t addressed the most important comments of the reviewer. So it still needs a major revision. The comments are as follows:

  1. Please report gel content of all samples.

The authors just mentioned the weight percent of PBS and LPBS in the revised manuscript!!! You have to measure the gel content for the samples and then report the values and justify the observed trend.

  1. Introduction is poor and it has to be improved by a thorough clarification of the prior works and related innovations as well as citing more references.

The authors just added a few refs! They have to consider the above comment to improve the quality of introduction.

  1. Fig 1 c: Please report all rheological data including complex viscosity and loss modulus curves for the samples (LPBS with different contents as well as PBS with different molecular weights).

Fig. is the curves of PBS with different molecular weights. The authors need to add curves for LPBS with different contents as mentioned in the comment and explain the trends.

-The English is still full of mistakes. The manuscript needs to be rewritten by a native speaker!

Author Response

Reviewer: 1

Comment 1:

 “The authors haven’t addressed the most important comments of the reviewer. So it still needs a major revision. The comments are as follows:”

Response 1:

We acknowledge the reviewer’s comments.

Comment 2:

“Please report gel content of all samples. The authors just mentioned the weight percent of PBS and LPBS in the revised manuscript!!! You have to measure the gel content for the samples and then report the values and justify the observed trend.”

Response 2:

The reviewer’s suggestion is significant.

In our experiment, the gel is present in the silicone rubber of PBS gel/MVQ composite. Since the structure of the gel is similar to that of silicone rubber, the gel is difficult to dissolve into a solvent and separate it. Moreover, the borate ester bond of Si-O-B formed by PBS gel is dynamic and reversible, and can only be partially dissolved under the action of solvent. It will be formed again when the solvent volatilizes, so it is difficult to completely remove the PBS gel in silicone rubber. Therefore, we used gel addition amount as gel content in silicone rubber to observe its influence on the damping performance of rubber.

Comment 3:

“Introduction is poor and it has to be improved by a thorough clarification of the prior works and related innovations as well as citing more references. The authors just added a few refs! They have to consider the above comment to improve the quality of introduction.”

Response 3:

  The reviewer’s suggestion is significant. According to the reviewer’s suggestion, the Introduction has been rewritten in the revised manuscript.

On Page 1, in the Introduction:

Polymer damping material is a kind of functional material which can convert mechanical vibration into heat dissipation [1,2]. This kind of material can be used to reduce the vibration and noise of all kinds of machinery and improve the precision and life of machinery. It has been widely used in the fields of transportation, municipal engineering, high-rise buildings, precision instruments, aerospace, military equipment and so on [3,4]. In fact, the effective high damping materials require a damping coefficient of tan δ>0.3 at wide temperature range [5]. However, the effective damping temperature of a general viscoelastic material is mostly near the glass transition temperature (Tg) and the temperature range is narrow [6]. With the increase of temperature, the resonance intensity increases, and its damping coefficient (tanδ) is significantly reduced, which makes it impossible to adapt some special working conditions, such as extreme conditions of high and low temperature alternating [7].

Methyl vinyl silicone rubber (MVQ) is long straight chain polysiloxane with high molecular weight. Its main chain is composed of Si-O-Si bonds, and its side groups are composed of organic groups. Due to the typical semi-organic and semi-inorganic structure, silicone rubber has both the thermal stability of inorganic polymer, and flexibility of organic polymer. Therefore, silicone rubber has excellent low temperature resistance and resistance to high temperature performance [8,9]. The glass transition temperature of silicone rubber is relatively low (-70~-110 °C), and the structure of Si-O-Si bond makes its mechanical properties keeping stable in a wide temperature range (-50~200 °C) [10]. The damping property of silicone rubber is mainly due to the dynamic deformation under the action of dynamic stress. The external force response of rubber is divided into elastic part, viscous part, and the strain will fall behind the stress. Importantly, the mutual friction between the molecules occurs when the rubber exhibits a cyclical change of stretching and retraction. Moreover, the mechanical energy is dissipated by the running of thermal energy, thereby achieving the effect of vibration and noise reduction [11,12]. However, the damping properties of silicone rubber are lower at room temperature, generally near the Tg (-129~-70 °C), and the tanδ of the material is generally below 0.1. It is necessary to modify the damping property to meet the requirements [13,14].

The commonly used methods include blending method, mutual transfer and network methods, and copolymerization method [15,16]. Adding additives is one of the effective methods to improve the damping properties of silicone rubber. A patent of Kobayashi [17] exhibited a way to enhance the damping properties by adding a kind of silicone powder and a surface-hydrophobic silica powder into the rubber, and finally, an excellent vibration damping performance and long-term storage stable rubber was obtained. And Quan-Ji S [18] provided a method to prepare high damping silicone rubber rom low phenyl silicone rubber, fumed silica, and self-synthesized damping agent, the rubber had an excellent damping property in 0~70 °C.

Polyborosiloxanes (PBS),which was invented initially in the search for substitutes of natural rubbers, possess reversible physical cross-links. A material like PBS may be denoted a supramolecular elastomer [19,20]. In all commercial applications, PBS constitutes the viscoelastic matrix, and inorganic/organic fillers are added for different engineering applications or lowering cost [21]. At room temperature, pure PBSs behave elastically under a rapid strain variation, and suffer from brittle fractures. However, on longer time-scales, they flow as a viscous fluid. The fascinating viscoelastic properties made PBSs applicable in education on various deformation processes [22]. The molecular structure of the silicone rubber PBS-gel is similar, and the shear hardening gel itself has a strong energy dissipation effect [23,24]. In our present study, the PBS-gel is blended with silicone rubber to prepare composite silicone rubber. The effects of shear hardening gel on the damping properties of silicone rubber are also investigated.

The references were adjusted, eight new references were added, and the original references [13], [14], and [16 ]were replaced and deleted in the revised manuscript.

Ref. 8 Thermochim. Acta. 2012, 529. 25-28.

Ref. 9 Polym. Degrad. Stab. 1993, 41, 109-116.

Ref. 13 J. Appl. Polym. Sci. 2002, 85, 746-751.

Ref. 14 J. Appl. Polym. Sci. 2011, 119, 2737-2741.

Ref. 17. U.S. Patent No 6,498,211, 2002.

Ref. 18. Nat. Mater. 2011, 10, 14-27.

Ref. 19. Nature 2008, 451, 977-980.

Comment 4:

“Fig 1 c: Please report all rheological data including complex viscosity and loss modulus curves for the samples (LPBS with different contents as well as PBS with different molecular weights). Fig. is the curves of PBS with different molecular weights. The authors need to add curves for LPBS with different contents as mentioned in the comment and explain the trends.”

Response 4:

The reviewer’s suggestion is significant.

  Fig 2a shows the loss modulus (G’) of LPBS, PBS and HPBS samples first increases and then decreases with the increase of frequency. This is because the relaxation phenomenon of the gel molecules makes the gel appear to be in a fluid state when the low-frequency shear occurs, so the loss modulus is high. With the increase of shear frequency, the gel gradually becomes solid and the loss modulus decreases. The complex viscosity is exhibited in Fig 2b,it decreases with the increase of shear frequency.

  In addition, the notes of Figure 2 have been revised in the revised manuscript.

Figure 2. (a) Loss modulus (G’) of LPBS, PBS and HPBS samples at different frequencies; (b) complex viscosity of LPBS, PBS and HPBS samples at different frequencies.

Comment 5:

“The English is still full of mistakes. The manuscript needs to be rewritten by a native speaker!”

Response 12:

  The reviewer’s suggestion is significant.

  The English syntax has been revised again in the revised manuscript.

Reviewer 2 Report

In my opinion this paper still needs major revision.

My detailed comments in the attachment.

Author Response

Reviewer: 2

Comment 1:

In page 1, line 12, this part is still not prepared according guidance for authros (see Polymers manuscript template on website: https://www.mdpi.com/journal/polymers/instructions).

Response 1:

  The corrected information has been revised in the revised manuscript.

Comment 2:

In Section 2.1 Materials, Mw instead of Mw, 9.14×104 g·mol-1 instead of 91.4 kg·mol-1, please provide some characteristics for these two rubber e.g. Mooney viscosity, density, tensile strength, hardness, etc.

Response 2:

The reviewer’s suggestion and question are significant.

  The viscosity of LPBS, PBS HPBS is 400,5000 and 50000 mPa·s respectively.

The methyl vinyl silicone rubber (MVQ) of RBB2400-30 were bought from Dow Corning, its tensile strength is 9.0MPa, and the elongation at break is 750%, the tear strength is 25.0KN/m, and its shore A hardness is 36.

Comment 3:

In page 5, line 148, please present the sol fraction value extracted by THF.

Response 3:

Although PBS can dissolve in THF, its solubility is not high. Since the borate ester bond of Si-O-B formed by PBS gel is dynamic and reversible, and can only be partially dissolved under the action of THF. It will be formed again immediately when the solvent volatilizes, so it is difficult to completely remove the PBS gel in silicone rubber to get the solvent fraction.

Comment 4:

In page 7, line 209, confirm this by sol fraction measurements.

Response 4:

  Due to the existence of Si-O-Si dynamic covalent bond, it is difficult to directly measure the gel fraction and gel content of PBS/MVQ and the interaction between molecules.Due to the interaction of long-chain molecules, we consider that the molecules of PBS gel and organosilicon chains will become entangled and form physical cross-links.

Comment 5:

In Table 2, Check values of elongation at break Value above 1000% are rather impossible add standard deviation to EB and hardness.

Response 5:

Comment 6:

In page 11, line 296-300, “The PBS-gel has excellent damping performance and similar to the molecular structure of silicone rubber, which can achieve good mixing and greatly enhance the damping performance of silicone rubber, especially maintain a high loss factor in a wide temperature range, when the contents of LPBS gel reached above 30 wt%, the loss factor of silicone rubber can be more than 0.5, and its effective temperature range from -20-125 °C.” This sentence is too long.

Response 6:

  Yes, the long sentence has been revised in the revised manuscript.

Reviewer 3 Report

I think the current revised version seems OK for me.

Author Response

Thank you very much for your suggestions and recognition.

Reviewer 4 Report

Manuscript has been significantly improved

Author Response

(The authors gave the same response as above.)

Round 3

Reviewer 1 Report

The authors have addressed the comments, so I recommend publication of the work entitled “Study on Optimization of Damping Performance and Damping Temperature Range of Silicone Rubber by  Polyborosiloxane Gel” in polymers.

Author Response

Thank you very much for your guidance and approval. The title of the paper has been modified based on your suggestion.

Reviewer 2 Report

Dear Authors,

Most of my comments were included in revised manuscript. However, I cannot find response on my comment:

Comment 5:

In Table 2, Check values of elongation at break Value above 1000% are rather impossible add standard deviation to EB and hardness.

Please confirm presented results and add standard deviation. After this modification paper can be considered for publication.

Author Response

The comments given by the reviewers are clear and concise, so I made corrections directly in the revised version based on his comments and marked them in the text.

Response 5:

The main reason why the elongation at break of PBS/MVQ composite silicone rubber exceeds 1000% is the effect of PBS gel. During the tensile test, the gel stress relaxation was obvious. In addition to molecular entanglement and Si-O-Si dynamic covalent bond and hydrogen bond, covalent cross-linking would not occur under the action of BPO. Therefore, the gel would increase the movement of polydimethylsiloxane molecules, causing the rubber fracture at the place of large strain.